# Reward signalling in brainstem nuclei under fluctuating blood glucose

**Tobias Morville[1], Kristoffer H. Madsen[2], Hartwig R. Siebner[1,3], Oliver J. Hulme** **[1]\***

**1** Danish Research Centre for Magnetic Resonance, Centre for Functional and Diagnostic Imaging and Research, Copenhagen University Hospital Hvidovre, Hvidovre, Denmark, **2** DTU Compute, Department of Informatics and Mathematical Modelling, Technical University of Denmark, Copenhagen, Denmark, **3** Department of Neurology, Copenhagen University Hospital Bispebjerg, Copenhagen, Denmark

\* oliverh@drcmr.dk

**Data Availability Statement:** A fully anonymized dataset along with analysis code for this paper is available at https://doi.org/10.5281/zenodo.4616695.

**Funding:** This work was supported by the following funders: H.R.S (Lundbeck Foundation Grant

## Abstract

Phasic dopamine release from mid-brain dopaminergic neurons is thought to signal errors of reward prediction (RPE). If reward maximisation is to maintain homeostasis, then the value of primary rewards should be coupled to the homeostatic errors they remediate. This leads to the prediction that RPE signals should be configured as a function of homeostatic state and thus diminish with the attenuation of homeostatic error. To test this hypothesis, we collected a large volume of functional MRI data from five human volunteers on four separate days. After fasting for 12 hours, subjects consumed preloads that differed in glucose concentration. Participants then underwent a Pavlovian cue-conditioning paradigm in which the colour of a fixation-cross was stochastically associated with the delivery of water or glucose via a gustometer. This design afforded computation of RPE separately for better- and worse-than expected outcomes during ascending and descending trajectories of serum glucose fluctuations. In the parabrachial nuclei, regional activity coding positive RPEs scaled positively with serum glucose for both ascending and descending glucose levels. The ventral tegmental area and substantia nigra became more sensitive to negative RPEs when glucose levels were ascending. Together, the results suggest that RPE signals in key brainstem structures are modulated by homeostatic trajectories of naturally occurring glycaemic flux, revealing a tight interplay between homeostatic state and the neural encoding of primary reward in the human brain.

## Introduction

A basic assumption of many models of adaptive behavior, is that the value of primary rewards are modulated by their capacity to rectify future homeostatic deficits [1, 2]. Compatible with this notion, deprivation-induced hypoglycaemia increases willingness to work for food in rats and humans [3], as well as the subjectively reported pleasure [2]. Dopamine is a neurotransmitter that plays a key role in signalling reward [4] and is involved in behavioural reinforcement, learning, and motivation [5, 6]. Via meso-cortical and mesolimbic dopaminergic projections, synaptic dopamine release modulates the plasticity of cortico-striatal networks and thereby sculpts behavioural policies according to their reward contingencies [4, 7].

ofExcellence "ContAct" ref: R59 A5399; Novo Nordisk FoundationInterdisciplinary Synergy Programme Grant "BASICS" ref: NNF14OC0011413) O.J.H(Lundbeck Foundation, ref: R140-2013-13057; Danish Research Council ref:12-126925) T.M (Lundbeck Foundation ref: R140-2013-13057).

**Competing interests:** The authors declare no competing interests. H.R.S. has received honoraria as speaker from Genzyme, Denmark and as senior editor of Neuroimage from Elsevier Publishers, Amsterdam, The Netherlands. H.R.S. has received a research fund from Biogen-idec, Denmark. This does not alter our adherence to PLOS ONE policies on sharing data and materials.

Patterns of phasic dopaminergic firing have been demonstrated to follow closely the principles of reinforcement learning, encoding the errors in the prediction of reward [6, 8–10]. Reward prediction error (RPE) signals appear commensurate with the economic construct of marginal utility, defined as the additional utility obtained through additional units of consumption, where utility is a subjective value inferred from choice [7, 11, 12].

Although animals are motivated by a homeostatic deficit of thirst or hunger, homeostatic states are rarely considered as relevant modulators of dopaminergic signalling of reward prediction errors. In typical paradigms involving cumulative consumption, the homeostatic deficit gradually diminishes as the animal plays for consumption of water or sugar-containing juice. Eventually, the animal rejects further play, presumably because the marginal utility of consumption diminished to a point of indifference or even aversion. Interestingly, a recent electrophysiology study in rats, demonstrated that oral consumption of sodium solution causes phasic dopaminergic signals in the nucleus accumbens, that are modulated by sodium depletion [13].

There is now growing evidence for a multifaceted interface between dopamine mediated reward-signalling and the systems underpinning energy homeostasis. Firstly, dopamine neurons in the ventral tegmental area (VTA) express a suite of receptors targeted by energy-reporting hormones ghrelin, insulin, amylin, leptin and Glucagon Like Peptide 1 (GLP-1) [14, 15]. This provides numerous degrees of freedom for flexibly interfacing between homeostatic state and reward signalling. Although hormonal modulations of phasic dopamine are yet to be fully scrutinised, there is emerging evidence that circulating factors do indeed modulate its magnitude. For instance, amylin, a hormone co-released with insulin, acts on the VTA to reduce phasic dopamine release in its mesolimbic projection sites [16]. In terms of neuronal input, there are many such opportunities for the appetitive control of dopamine mediated signalling.

Appetitive control can be delineated into three interacting systems [17]. The first system generates a negative valence signal which involves activity of the Agouti-related peptide (AgRP) neurons of the arcuate nucleus of the hypothalamus (ARC). Activity of $ARC_{AgRP}$ neurons reports on energy deficits, inhibits energy expenditure, and regulates glucose metabolism [18–21]. ARC neurons that contain peptide products of pro-opiomelanocortin (POMC) form an opponent code compared with $ARC_{AgRP}$ neurons. The balance between the two neuronal ARC sub-populations putatively encodes the value of near-term energetic states, becoming rapidly modulated just prior to food consumption [22]. The second system codes positive valence signals and consists of circuits involving the lateral hypothalamus (LH). It is linked to positively reinforcing consummatory behaviours via its GABAergic projections to VTA dopamine neurons [23] assumed to trigger positive feedback to keep consumption going during feeding bouts. The third valuation system involves calcitonin gene-related protein (CGRP)-expressing neurons in the parabrachial nuclei (PBN) that potently suppress eating when activated, but do not increase food intake when inhibited. $PBN_{CGRP}$ neurons are activated by signals associated with food intake, and they provide a signal of satiety that has negative valence when strongly activated [24]. The PBN has been characterised as a hedonic hotspot, the modulation of which by either GABA or Benzodiazepines potently modulates experienced reward [25]; $ARC_{AgRP}$ neurons GABA-ergically inhibit PBN neurons, thus stimuli predicting glucose consumption should inhibit $ARC_{AgRP}$, releasing the PBN from inhibition [26]. Further, hormones related to hunger and feeding (GLP-1 & leptin) modulate PBN activity and subsequent behaviour [27, 28]. Of note, these three valuation systems all project to and modulate the dopaminergic neurons in the ventral tegmental area ($VTA_{DA}$). The interface between these hypothalamic-brainstem networks and the $VTA_{DA}$, is arguably the most important interface for mediating the dialogue between energy homeostasis and value computation.

While most evidence for encoding of RPEs is obtained under homeostatic deprivation, the modulation of RPE signalling triggered by physiological fluctuations in glucose availability (glycaemic flux) remains yet to be characterised in the human brain. This begs the questions, how are RPE signals modulated by these subcortical circuits that integrate, evaluate, and predict energy-homeostatic states? We hypothesize that glucose fluctuations above and below average levels of serum glucose, will down and up modulate RPE responses in these hypothalamic-brainstem networks. To test these hypotheses, we acquired a large volume of fMRI data in five participants during a simple Pavlovian cue-conditioning task, while their serum glucose was systematically manipulated.

## Methods

### Subjects

Five healthy (3 male), normal-weight subjects in the age range 23 to 29, participated in the study. Exclusion criteria were: $20 > BMI > 25$; $18 > Age > 32$ yrs; any metabolic or endocrine diseases or gastrointestinal disorder; any known medication that might interfere with the study; claustrophobia; and any metal implants or devices that could not be removed. Informed consent was obtained in writing from all subjects as approved by the Regional Ethics Committee of Region Hovedstaden (protocol H-4-2013-100) and in accordance with the declaration of Helsinki.

### Experimental procedure

The experimental design constituted a single-blinded, randomised control trial, with repeated measures crossover-design. On four separate days, subjects fasted for a minimum of twelve hours before testing. Compliance with the fasting instruction was based both on trust, and on the understanding that we would be able to detect if participants had not fasted via blood tests. Any participant that was not in the hypoglycemic range (defined here as <6mmol/L) at the start of the experiment would be assumed to have not fasted, and the session would be aborted. This was not necessary for any participant. At the beginning of an experimental session, subjects ingested either a hi-glucose (75 g, 300 kcal) or lo-glucose preload (10 g, 40 kcal) diluted to 100 ml with a non-caloric lemon juice, used in order to mask the taste of the glucose. The lo-glucose preload resulted in glucose ascending during the fMRI acquisition period, due to the consumption of glucose, whereas during the same period after the hi-glucose preload, glucose levels descended (Fig 1B). The preload conditions are thus referred to as ascending and descending conditions. The temporal order of the conditions was randomised within subjects, with each condition being performed twice. Both preloads were anecdotally reported by independent samplers to be highly palatable. Each delivery of glucose reward was 0.4ml, corresponding to 0.3 g of glucose (1.2 kcal) per delivery.

### Experimental task

After consuming the preload, participants engaged in a simple Pavlovian cue-conditioning task. The colour of the fixation cross cued both the onset of each trial ($Cue_{onset}$), as well as stochastically predicting glucose delivery (Fig 1A), with one colour signalling a high probability of glucose delivery ($Cue_{high}$), and another signalling a low probability ($Cue_{low}$). 10–15 seconds after delivery of the liquid, a purple cross signalled that subjects were to swallow. The large temporal distance between the swallowing and the reward onset, as well as the levels of temporal jitter, was designed to mitigate the contamination of the reward signals by swallowing related artefacts. All probabilities and contingencies were implicitly revealed only through

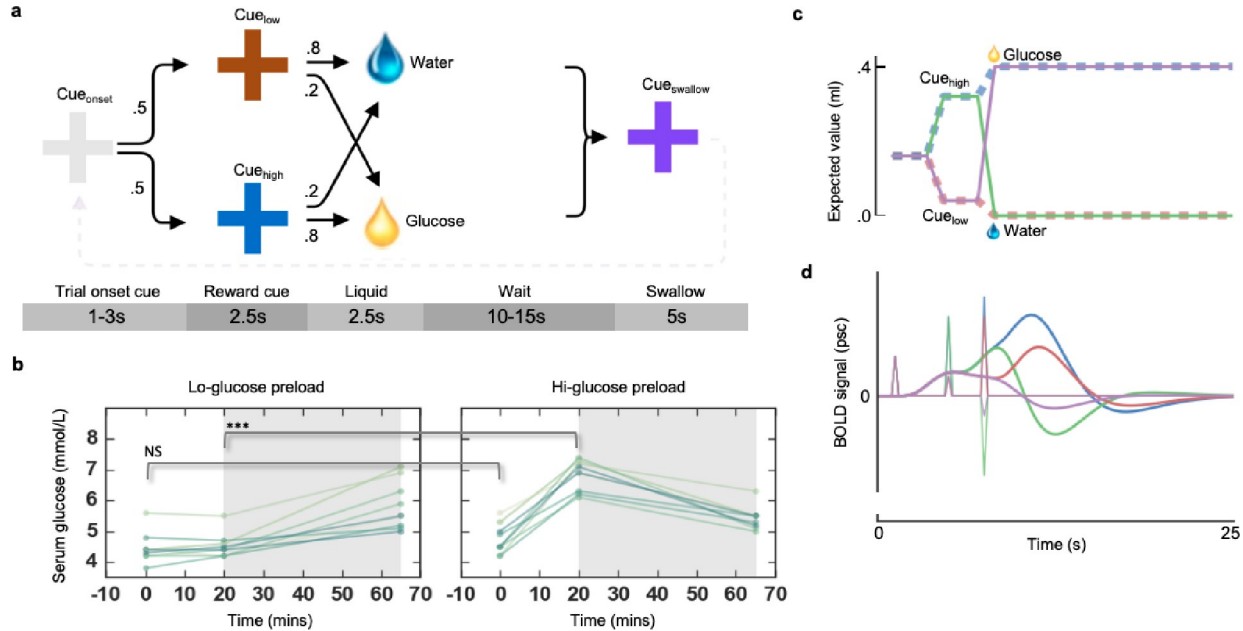

**Fig 1. Experimental design, glucose trajectories, and expected reward signals.** a, participants were presented with the Cue$_{onset}$ (grey fixation cross) for 1-3s after which either Cue$_{high}$ (blue cross) or Cue$_{low}$ (brown cross) is presented with a probability of 0.5 each. Cue$_{high}$ signalled a high probability (0.8) of glucose delivery and a low probability (0.2) of water delivery. Cue$_{low}$ signalled a low probability of glucose (0.2) and a high probability of water (0.8). The 0.4ml of the liquid were delivered over 2.5 seconds, followed by 10-15s wait period and a Cue$_{swallow}$ that cued the subject to swallow (here, purple) which lasted for 5s. All jitters are uniformly distributed within the ranges specified. b, serum glucose trajectories for the high and low glucose preload conditions. Grey shading indicates the period of fMRI acquisition for a single session. The different line plots indicate different sessions for all subjects. Glucose levels ascend during the fMRI acquisition period in the lo-glucose condition, and descend in the hi-glucose condition. c, graph depicts the objective reward expectations, expressed as the expected value in ml glucose, and the perturbation of these expectations under the onset of the experimental cues and outcomes. Note that reward expectations are updated three times per trial: at the onset of the Cue$_{onset}$; at the onset of Cue$_{high}$ or Cue$_{low}$; at the onset of Outcome$_{glucose}$ or Outcome$_{water}$. d, illustrates simulated BOLD responses to RPE signals resulting from the updated reward expectations shown in c, generated by convolving the canonical hemodynamic response function with the RPE stick functions evoked by changes to the reward expectations.

experience in the scanner, and all were stationary over all test days. The mapping between colour and outcome probabilities was counterbalanced across subjects, while the mapping was stationary within and between sessions. Participants went through ~82 trials [82 ± 1.5 SEM] each day giving ~328 trials per subject. Serum glucose measurements were attained immediately before and 20 minutes after ingestion, using a Contour® Next glucose meter (Fig 1B). It should be noted that whilst there is no jitter between the cue and the outcome, independent estimation of each of the effects is achievable by virtue of their probabilistic transitions (Fig 1A).

## Scanning procedure

Task related changes in regional brain activity were mapped with blood oxygen dependent (BOLD) MRI immediately after the second glucose measurement (t20). Functional MRI measurements were performed with a 3T Philips Achieva and a 32 channel receive head coil using a gradient echo T2* weighted echo-planar image (EPI) sequence with a repetition time of 2526 ms, and a flip-angle of 80°. Each volume consisted of 40 axial slices of 3 mm thickness and 3 mm in-plane resolution (220 x 220 mm). The axial field-of-view was 120 mm covering the whole brain, cutting off the medulla oblongata partially. This sequence was extensively piloted and optimised specifically for reducing distortion and maximising resolution with the hypothalamic and brainstem regions of interest. During each session, 800 EPI volumes were

acquired, resulting in 3200 EPI volumes per subject. Further, an anatomical T1-weighted image was recorded for each subject. Respiration and heart rate were measured to assess and model possible artefacts. Liquid tastants were contained in two 50 ml syringes, one containing water-only (water hence) the other containing glucose and water (glucose hence) solutions, attached to two programmable syringe pumps (AL1000-220, World Precision Instruments Ltd, Stevenage, UK), controlled by the stimulus paradigm script. The liquid was delivered orally via two separate 5m long 3mm wide silicone tubes. Each tube was attached to a gustatory manifold specifically built for the Philips head-coil (John B. Pierce Laboratory, Yale University). Visual stimuli were presented on a screen positioned ~30 cm away from the scanner.

## Pre-processing

Pre-processing and image analysis were done using SPM12 software (Statistical Parametric Mapping, Wellcome Department of Imaging Neuroscience, Institute of Neurology, London, UK). To correct for motion, EPI scans were realigned to their mean using a two-step procedure and co-registered to the T1 weighted anatomical image. The realigned images were spatially normalised to the standard ICBM space template of European brains, with a resampled voxel size of 3 mm.

## fMRI analysis

After model specification, the ascending condition sessions were concatenated using the function spm_fmri_concatenation (SPM 12) for each subject. Equivalently, the same concatenation was performed for the descending conditions sessions. A first-level fixed effects model was run over all subjects. The concatenation was performed to avoid state-dependent effects being expressed via the session-specific regression coefficients. All variables of interest were convolved with the canonical hemodynamic response function, along with their associated temporal and dispersion derivatives and fitted to the data using the specified GLM. The temporal evolution of cues and outcomes were modelled as separate conditions, each with state as parametric modulators. Regressors of no interest included a discrete cosine transform based 1/128 Hz cut-off frequency high-pass filter, rigid body realignment parameters using a 24 parameter Volterra expansion [29] and physiological noise from heart rate and respiration using the RETROICOR method [30]. We specified the striatum (caudate, putamen and nucleus accumbens), brainstem (pons, ventral tegmental area and substantia nigra) and hypothalamus as regions of interest (ROI). These ROIs were determined on the basis of the literature describing dopamine projections from midbrain to the striatum and its role in regulating behaviour as a function of reward. The pons was selected to accommodate the literature described above, which sets certain nuclei within the pons as important homeostatic modulators. All ROI were defined with the WFU pick atlas [31, 32] and cross checked against the book *Atlas of the human brain* [33]. All initial first-level analysis was performed as whole-brain uncorrected at $p < 0.001$. Significant clusters in regions of interest (ROI) are all reported as small-volume corrected with a family-wise threshold of $p < 0.05$ at cluster level (abbreviated SVC FWE), unless otherwise stated.

## Modelling RPEs

At the first level, a general linear model (GLM) was set up to model cue and outcome related brain activity. We specified separate regressors which modelled the onset of cue events (Cue$_{onset}$, Cue$_{high}$ and Cue$_{low}$) and outcome events (Outcome$_{gluc}$ & Outcome$_{water}$). Since there is no overt behavior in this task to which a temporal difference learning algorithm can be fit, we used a different approach based on how RPEs converge to changes in the conditional

expectation values of reward outcomes. As subjects learn the contingencies between the cues and the outcomes, the RPEs evoked by these events converge toward the change in expected value of the reward (here the volume of glucose), conditional on the events they have cumulatively experienced during the trial. This has been shown in single cell recordings, where the RPE signals of dopaminergic cells in the VTA, signal errors whose magnitudes reflect the change in expected value of juice volume, conditional on the events that have been experienced at that time [11]. Fig 1C illustrates how the expectation value of glucose volume evolves over time as a function of the cues observed. The conditional expectation value upon seeing the $Cue_{onset}$ is the expectation value of glucose volume for each trial, conditional on the fact the trial has started. This is shown in Fig 1C, since all lines begin from this starting point of 0.2ml. Thus, $Cue_{onset}$ triggers a small positive RPE, seen as the first spike on the left in Fig 1D. From there the expectation value of glucose increases or decreases depending on whether the $Cue_{high}$ or $Cue_{low}$ is experienced, as labelled in Fig 1C. This causes the second set of spikes seen in Fig 1D, where the green corresponds to the $Cue_{high}$ and the purple corresponds to $Cue_{low}$. Finally, the outcomes arrive changing again the conditional expectational value of glucose to either 0.4ml or 0ml. This can be thought of as the expectation value of the glucose that the agent can expect to metabolise having received the liquid in its mouth. This corresponds to the third set of spikes in Fig 1D. To approximate the RPEs without behavior, we specified contrasts which were formulated by linear combinations of these cue and outcome regressors, weighted as a function of the RPE values that would be expected from the temporal-difference learning algorithm, once converged [34]. In other words, the cue and outcome regressors were weighted by the change in conditional expectation of glucose volume caused by the regressor's event (either the cue or the outcome). A contrast of positive RPE signals ($RPE_{pos}$) was computed by assigning the positive valence cue and outcome regressors (i.e. $Cue_{onset}$, $Cue_{high}$, and $Outcome_{gluc}$) contrast weights that were proportional to the change in expectation value of glucose volume that these events caused. Equivalently negative RPEs ($RPE_{neg}$) were computed equivalently as contrasts to include only the negative valence events (i.e. $Cue_{low}$ and $Cue_{water}$). It should be noted here that in this approach to modelling RPEs, the model does not incorporate any effect of learning, in effect modelling what signals are expected once learning has converged on the expected reward values.

## Modelling RPE modulation by glucose

The effect of serum glucose on RPE was modelled via first order parametric modulator of the cue and outcome regressors, taking the interpolated serum glucose at each time point as the covariate (demeaned). This resulted in five parametric modulator regressors, namely $Cue_{onset*state}$, $Cue_{high*state}$, $Cue_{low*state}$, $Outcome_{gluc*state}$ & $Outcome_{water*state}$. From these regressors, contrasts can be specified to model the effect of serum glucose on RPEs. The effect of glucose state on positive RPEs ($RPE_{pos*state}$) was computed as a linear combination of $Cue_{onset*state}$, $Cue_{high*state}$, and $Outcome_{gluc*state}$ regressors. Equivalently, the effect of glucose state on negative RPEs ($RPE_{neg*state}$) was computed as a linear combination of ($Cue_{low*state}$, $Outcome_{water*state}$). All 1st-order parametric modulators are orthogonal to their associated onset regressors by construction. No other orthogonalization of regressors was performed.

## Results

### $Cue_{onset}$ induced brain activity

An RPE signal should respond to the $Cue_{onset}$, with an error signal that signals the expected value of glucose reward for the whole trial [11]. Computing the main effect of this regressor, this was found to evoke an increase in activity in VTA bilaterally (Fig 2A). Thus cue-induced

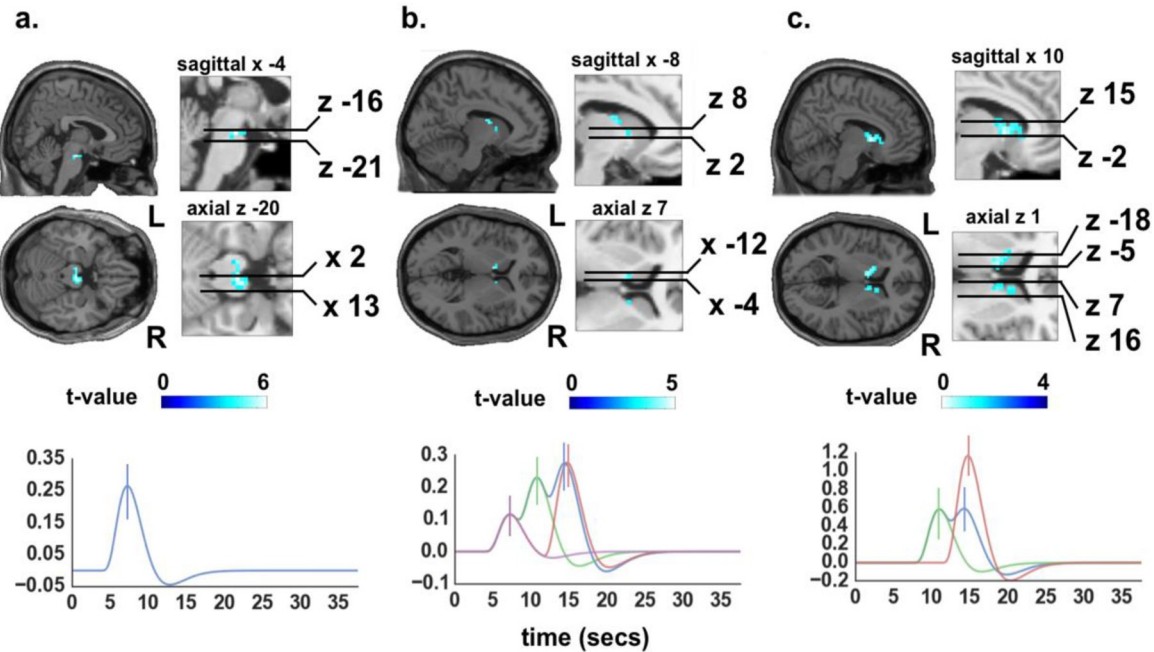

**Fig 2. Statistical parametric maps of main effects of trial onset, positive and negative RPEs.** a, main effect of $Cue_{onset}$, which reflects an RPE following the mean reward expectation for the whole trial, revealed activity in VTA bilaterally ($\beta$ = 2.77) (R: [4–17–20] and L: [–8–17–20], FWE SVC). Further this revealed deactivation of precentral gyrus (primary somatosensory cortex), mediodorsal thalamus, and striatum (FWE whole brain, not shown). Lower panel shows fitted response for the $Cue_{onset}$ event within the same region. b, main effect of $RPE_{pos}$ revealed activity in left lateral caudate [$\beta$ = 1.21; coordinates -8 4 7; FWE SVC]. Lower panel shows fitted response for the $RPE_{pos}$ contrast within the region. The colours of time courses follow the same meaning as shown in Fig 1C and 1D. c, main effect of $RPE_{neg}$ revealed bilateral activity in caudate (L: -11–2 13; R: 10 7 1; $\beta$ = 12.2) medial dorsal thalamic nucleus [7, –2, 22], and lateral insula [43, –2, –17] (all FWE). Lower panel shows fitted response for the $RPE_{neg}$ contrast within the region. All fitted responses were generated by convolving the canonical hemodynamic response function with the RPE stick function multiplied by their respective beta-values extracted from the local maxima of the ROI in units of percent signal change. Again, the time courses follow the same meaning as shown in Fig 1C and 1D. Error bars show standard errors of the mean.

VTA activation is consistent with existing evidence of VTA signalling RPEs [35–37]. $Cue_{onset}$ also led to the deactivation of the postcentral gyrus (primary somatosensory cortex), mediodorsal thalamus, and the striatum [whole brain, uncorrected p < 0.001] (not shown).

## Positive and negative reward prediction error signals

In several brain regions, regional task-related activity changed in proportion with the magnitude of positive-going (i.e. better-than-expected) reward prediction errors ($RPE_{pos}$) or negative-going (i.e. worse-than-expected) reward prediction errors ($RPE_{neg}$). Task related activity scaling with the $RPE_{pos}$, formalized as an RPE-weighted linear combination of $Cue_{trial}$, $Cue_{high}$, and $Outcome_{gluc}$, was found in left lateral caudate nucleus (Fig 2B). Conversely, task related activity reflecting $RPE_{neg}$, formalized as an RPE-weighted linear combination of $Cue_{low}$ and $Outcome_{water}$, was located in the caudate nucleus bilaterally Fig 2C), the medial dorsal thalamic nucleus, and insula (not shown).

## Modulation of task-related brain activity by glycaemic state

We were interested to identify changes in RPE signalling over time as serum glucose either ascended or descended. A bilateral cluster, including the parabrachial nuclei (PBN), showed a modulation of the regional neural responses to RPEs by the glycaemic state dynamics (Fig 4A).

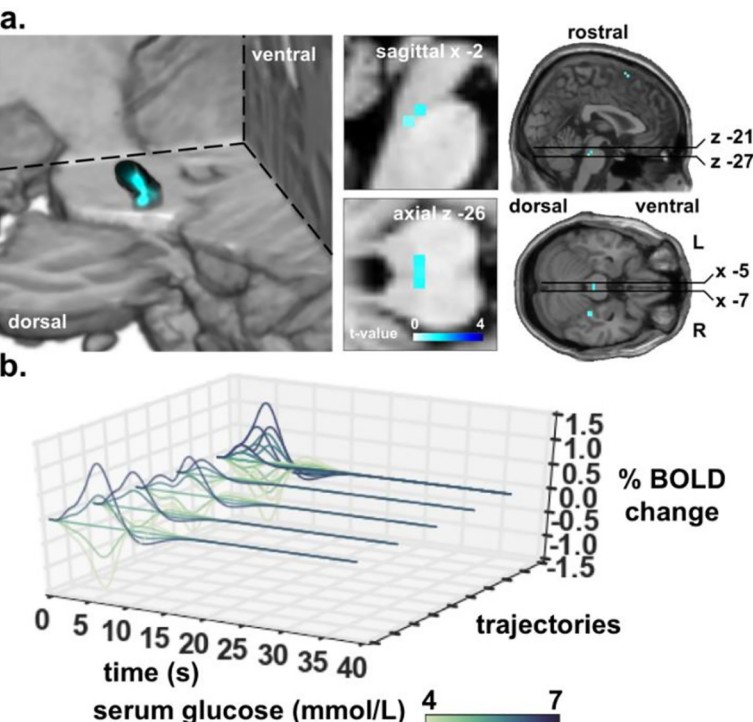

**Fig 3. Statistical parametric maps of RPE$_{pos*state}$ and fitted responses over varying glycaemic state.** a, main effect of RPE$_{pos*state}$ revealed bilateral activity in the PBN [-2–29–26; FWE SVC]. The colour bar indicates the t-value on a scale of white to blue. b, fitted response ($\beta$ = 1.66) of the local maxima of PBN cluster (7 voxels) to the possible trajectories that RPE$_{pos*state}$, yield (Fig 1D) modulated by serum glucose state. There are four possible trajectories of the positive RPE, according to the four different possible trial types depicted in Fig 1C. These trajectories are modulated by serum glucose and shown as different colours with 5 different equally spaced glycemic states. There are thus 4 sets of these trajectories (the four closest to the viewer), each showing their modulation by glucose in the different colours. In order of closeness to the viewer, the first set of trajectories is for the Cue$_{high}$- Outcome$_{gluc}$ trial; the second set is for the Cue$_{high}$−Outcome$_{water}$ trial; the third set is for Cuelow−Outcome$_{gluc}$ trial; the fourth set is for the Cuelow −Outcome$_{water}$ trial. The fifth set of trajectories (furthest from the viewer) superimpose together all possible trajectories depicting the complexity of how the trajectories are patterned according to their modulation by glucose state. For comparison, this presentation is analogous to the superimposed trajectories shown in Fig 1D. Note that the colour bar for the graphs in (b) is serum glucose and is distinct from the colour bar in (a) used to indicate t-values.

Higher levels of serum glucose amplified the response to RPE$_{pos}$ in the PBN region (Fig 3B). The main effect of RPE$_{neg*state}$, which models the interaction between RPE$_{neg}$ and state, did not yield any significant results in any ROI, or in exploratory analyses using uncorrected thresholds, in positive or negative contrasts. When considering both ascending and descending serum glucose fluctuations together, there was no detectable region where the RPE$_{neg}$ signal was either positively or negatively modulated by serum glucose. Brain responses to Cue$_{onset}$ were also not altered by glycaemic state.

We also tested for state-dependent modulatory effects on RPE processing which depends on whether serum glucose was ascending (Fig 1B, left) or descending (Fig 1B, right) over time. This yields four different contrasts (ascending vs. descending and the converse, for RPE$_{pos*state}$ and RPE$_{neg*state}$) that are directly relevant to glucose state. Subtracting descending trajectories from ascending and vice versa, revealed no significant activity changes for RPE$_{pos*state}$ [whole brain, uncorrected]. The same comparisons for RPE$_{neg*state}$ did reveal significant effects in VTA and substantia nigra for ascending trajectories relative to descending trajectories (Fig 4A). This result shows a relative amplification of the RPE$_{neg*state}$ signal as glucose state increases. In instances where reward was lower-than-expected (thus yielding negative RPE),

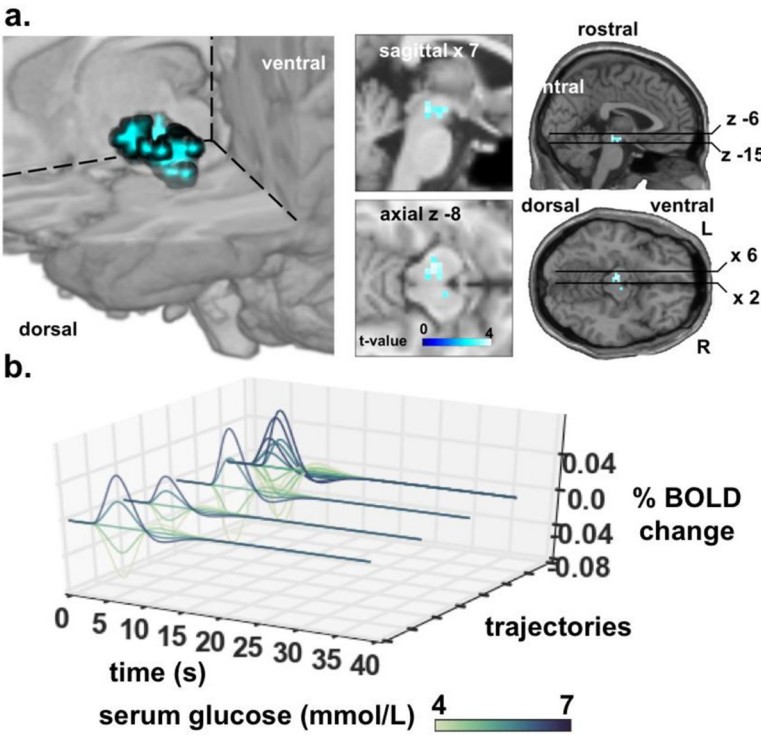

**Fig 4. Statistical parametric maps of RPE$_{neg*state}$ subtracted for increasing minus decreasing.** a, negative reward prediction error RPE$_{neg*state}$ revealed glucose modulated activity in SN [±12, -22, -10] and VTA [0, −15, −9] when subtracting the effect of descending from the ascending glucose state [FWE SVC]. b, fitted response ($\beta$ = 0.34) of the local maxima of cluster [7, -11, 8; 52 voxels] to the three possible trajectories that RPE$_{neg*state}$ yield modulated by serum glucose state. Onsets are not at zero because the negative trajectories do not envelop the trial mean which has a positive expectation. In order of closeness to the viewer, the first set of trajectories is for the Cue$_{high}$- Outcome$_{gluc}$ trial; the second set is for the Cue$_{high}$−Outcome$_{water}$ trial; the third set is for Cuelow−Outcome$_{gluc}$ trial; the fourth set is for the Cuelow−Outcome$_{water}$ trial. The fifth set of trajectories (furthest from the viewer) superimpose together all possible trajectories depicting the complexity of how the trajectories are patterned according to their modulation by glucose state. The set of trajectories furthest away from the viewer superimposes the trajectories from all trial types into one plot.

the glucose state modulated the RPE$_{neg}$ signal in VTA and SN more so when glucose levels were ascending than descending.

## Discussion

Participants performed a simple cue-conditioning task involving the probabilistic delivery of glucose or water, whilst their blood glucose fluctuated over the course of an hour. We had hypothesized that low levels of serum glucose will positively modulate the scale of positive RPE responses in hypothalamic-brainstem networks, reflecting the marginal utility of glucose as a function of homeostatic needs. Contrary to this hypothesis, we did not observe any positive RPE that increased its scaling with decreasing levels of serum glucose levels. In exploratory analyses there were however several observations of a dependency between serum glucose and RPE signals. Reward prediction error signalling in the parabrachial nuclei scaled positively with serum glucose levels, and this was true whether glucose was ascending or descending over time. We found that both the VTA and SN became more sensitive to negative RPEs for ascending compared to descending glycaemic trajectories. We begin by discussing the interpretation of these state modulated RPE effects, before considering other effects, and the limitations inherent under this paradigm.

In rodent models, the PBN acts as a $2^{nd}$ order relay of inputs from the nucleus tractus solitarius, and is critical in the control of energy homeostasis via its projections to amygdala [38, 39], VTA [40], hypothalamus [39, 41] and the nucleus accumbens [42]. Subnuclei of the PBN are targeted by descending projections from several nuclei implicated in energy homeostasis, including hypothalamus, amygdala, and the bed nucleus of the stria terminalis [39, 43]. The PBN is known to be a potent site of reward modulation and subsequent behaviour in rodents. Microinjection of benzodiazepines [44–46], endocannabinoids [47], opioids [48, 49] and melanocortin agonists [50] into the PBN, all evoke hyperphagias. To our knowledge, the involvement of PBN in context of hedonics and reward signalling in the human brain remains to be charted. Here we provide tentative evidence that PBN activity generates a positive RPE-like signal that is sensitive to blood glucose and is time-locked to both the sensory cues predicting glucose, as well as glucose consumption.

We found no state modulation of $RPE_{neg}$ signalling ($RPE_{neg*state}$), expressed during both ascending and descending glycaemic trajectories. For the $RPE_{neg}$ signal, the modulatory effect of the glycaemic trajectory depended on whether glucose trajectories were ascending or descending. Regional activity scaling with $RPE_{neg}$, the VTA and SN showed significantly higher state modulation effects during ascending compared to the descending glycaemic paths. In our experiment, the ascending glucose trajectory resulted from a low-glucose preload with the subsequent increase over time likely occurring by virtue of the continual ingestion of glucose throughout the paradigm (Fig 1B). In the ascending condition, the neural response to $RPE_{neg}$ is attenuated at lower levels of serum glucose, while it becomes amplified by the transition to higher serum glucose. Given that there is some evidence that dopaminergic neurons of the VTA and SN are directly inhibited by insulin [15], it is possible that the insulin release following hi-glucose preload was highest at the start of the paradigm, decreasing over time, and thus resulting in a gradual decrease in inhibition. However, it should be noted insulin can have a stimulatory effect on dopaminergic firing rates [51]. The difference in $RPE_{neg}$ in its state modulation ($RPE_{neg*state}$) between ascending and descending may therefore be attributed to differential dynamics of insulin secretion [52], though other hormones such as ghrelin [52–54] or leptin [55–58] may play a role. It is not presently known, why it would make sense that a behaviorally reinforcing signal, such as phasic RPE, is expressed less as glucose levels are decreasing in a situation where the body is moving towards a state of potential dyshomeostasis.

Our finding that the VTA and SN responses are linked to $RPE_{neg}$ may appear counterintuitive, given that these midbrain regions are typically associated with BOLD responses signalling positive-going RPEs. This is assumed to be by virtue of the fact that a greater range of firing rates can be devoted to the better-than-expected range, signalled by above baseline firing. This is contrasted to the worse-than-expected range, which can only be signalled by a decrease from an already low baseline frequency. It is conceivable that what we are asserting as being $RPE_{neg}$ is in fact a positive RPE resulting from the gradual avoidance of glucose, which increases in magnitude with increasing levels of serum glucose as reported in humans [2] and rats [59]. Thus, as the experimental paradigm continues, especially under the conditions of glucose preload, serum glucose increases, and this may change the valence of the outcome, switching the affective connotation of glucose from palatable to aversive.

As detailed in the introduction, little is known about how the interface between dopaminergic RPE signalling and energy homeostasis is implemented in the human brain. While there are many means by which circulating factors can modulate activity in the VTA and SN, the mechanisms by which this is mediated cannot be revealed without wider hormonal assays. Contemporaneous hormonal sampling, as well as continuous glucose monitoring in the scanner will prove an important step in revealing these latent factors.

There are several technical limitations that should be noted in discussing this experiment. Though relatively high volumes of functional data (150 minutes per subject) were acquired in each subject, the total number of subjects was small. The reason for this was a focusing on maximising experimental power within subjects, for finer scale inference on the longer time-scale glucose dynamics. It was not known ahead of time how large the modulatory effects would be, and thus we deployed a conservative strategy of testing fewer subjects for longer. The long regressors that result from the concatenation of sessions may have meant that the high pass filtering would have reduced the effects of ascending and descending glucose levels. Inferring slow timescale dynamics is generally a problem for fMRI, however it is circumvented to some degree here, insofar as we are inferring the modulatory effect of glucose on faster RPE signalling dynamics which occur on a faster timescale than that which is filtered by the high-pass filter. Due to the small number of subjects, we deployed a fixed effects analysis over all subjects. It should be noted that this makes assumptions about the nature of the noise that might not be compatible with the repeated measures design, since it is difficult to correct for non-sphericity in this setting. Future work will expand this paradigm with a larger group of subjects to afford random effects modelling, and thus generalisation to the population sampled from. Contrary to our hypotheses, we found no modulatory effect of hypothalamic nuclei on RPE signalling. We stress that the current imaging protocols and field-strength (3T) were not optimal to dissociate neural activity in the hypothalamic nuclei. Due to the proximity of air sinuses adjacent to the hypothalamus and the effective resolution available, the present study most likely had insufficient sensitivity to capture activity in hypothalamic regions of interest. Another issue is whether the hemodynamic modelling was appropriate for detected evoked responses in subcortical regions which may deviate from the canonical hemodynamic response functions typically used. The regression model we used deployed temporal and dispersion derivatives for all regressors of interest in order to account for idiosyncratic variance in the timing and temporal spread of the hemodynamic response function. Finally, it should be noted that the cue-conditioning employed in this study was passive. Hence, subjects produced no overt choice behaviour against which to fit learning rate parameters for the RPE model, instead we relied on the asymptote values for the RPE signals. The problem of modelling RPEs in the absence of choice behaviour, motivates fitting learning rate parameters directly to brain data, a computational imaging approach that future work will exploit [60].

Recent literature on the computational neuroscience of reinforcement Learning (RL) has evidenced how decision-making in the mammalian brain is driven by optimizing the net value of both primary and non-primary rewards. Such reward computations have been shown to rest on a comparison between the expectation and outcome of external environmental cues, integrating both the physical and cognitive effort costs of the agent [61–63]. The work presented here tentatively expands this perspective this by showing that reward and RPE signals are dependent on internal homeostatic states, which may serve to modify the motivational values according to the personal and time varying homeostatic needs of the organism.

In conclusion, we exploited a simple paradigm, capable of eliciting RPEs under differential glycaemic trajectories, to identify brain stem structures that show a modulation of RPE signalling depending on the glycaemic state. We found that the PBN signals a positive-going reward prediction that is subject to systematic modulation by serum glucose. In the VTA and SN, negative-going RPEs were modulated by serum glucose trajectories, but in a way that was specific to an ascending glycaemic slope. Together the results show that RPE signals in key brainstem structures can be modulated by homeostatic trajectories inherent in naturally occurring glycaemic flux, revealing a potentially tight interplay between homeostatic state and the signalling of primary reward in the human brain.

## Acknowledgments

We thank Mehdi Keramati and Boris Gutkin for several helpful discussions.

## Author Contributions

**Conceptualization:** Tobias Morville, Hartwig R. Siebner, Oliver J. Hulme.

**Data curation:** Oliver J. Hulme.

**Formal analysis:** Tobias Morville, Oliver J. Hulme.

**Funding acquisition:** Oliver J. Hulme.

**Investigation:** Tobias Morville, Oliver J. Hulme.

**Methodology:** Tobias Morville, Kristoffer H. Madsen, Oliver J. Hulme.

**Project administration:** Oliver J. Hulme.

**Resources:** Hartwig R. Siebner, Oliver J. Hulme.

**Supervision:** Hartwig R. Siebner, Oliver J. Hulme.

**Validation:** Tobias Morville, Kristoffer H. Madsen, Oliver J. Hulme.

**Visualization:** Tobias Morville, Oliver J. Hulme.

**Writing – original draft:** Tobias Morville, Oliver J. Hulme.

**Writing – review & editing:** Tobias Morville, Kristoffer H. Madsen, Hartwig R. Siebner, Oliver J. Hulme.

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
