## [Decision Letter · Decision Letter 0]

21 Jul 2020

PONE-D-20-05797

Reward signalling in brainstem nuclei under fluctuating blood glucose

PLOS ONE

Dear Dr. Hulme,

Thank you for submitting your manuscript to PLOS ONE. After careful consideration, we feel that it has merit but does not fully meet PLOS ONE’s publication criteria as it currently stands. Therefore, we invite you to submit a revised version of the manuscript that addresses the points raised during the review process.

The main comment is the lack of methodological detail (see comments of both reviewers). It is impossible to currently understand how the TD model was constructed. Note that a TD model requires adding assumptions that may change its qualitative behavior (Pan et al., 2005, JNeurosci), so it must be made explicit. Equally important, I could not reconstruct how the 1st-level GLM was constructed (which events? which parametric modulators? were they orthogonalized?), and how contrasts were subsequently defined based on the 1st-level regressors.

Relatedly, it seems that the modulation by serum glucose runs opposite to what the authors predicted (more glucose, stronger response). If so, this must be made explicit in the Discussion.

We look forward to receiving your revised manuscript.

Kind regards,

Tom Verguts

Academic Editor

PLOS ONE

Journal Requirements:

2. Please provide additional details regarding participant consent. In the Methods section, please ensure that you have specified what type of consent you obtained (for instance, written or verbal) and whether the ethics committee approved this consent procedure. If verbal consent was obtained please state why it was not possible to obtain written consent and how verbal consent was recorded. If your study included minors, state whether you obtained consent from parents or guardians.

3.We note that you have stated that you will provide repository information for your data at acceptance. Should your manuscript be accepted for publication, we will hold it until you provide the relevant accession numbers or DOIs necessary to access your data. If you wish to make changes to your Data Availability statement, please describe these changes in your cover letter and we will update your Data Availability statement to reflect the information you provide.

4.Thank you for stating the following in the Competing Interests section:

[The authors declare no competing interests.  H.R.S. has received honoraria as speaker from Genzyme, Denmark and as senior editor of Neuroimage from Elsevier Publishers, Amsterdam, The Netherlands. H.R.S. has received a research fund from Biogen-idec, Denmark.].

Reviewers' comments:

Reviewer's Responses to Questions

**Comments to the Author**

1. Is the manuscript technically sound, and do the data support the conclusions?

Reviewer #1: Yes

Reviewer #2: Partly

2. Has the statistical analysis been performed appropriately and rigorously? 

Reviewer #1: Yes

Reviewer #2: No

3. Have the authors made all data underlying the findings in their manuscript fully available?

Reviewer #1: Yes

Reviewer #2: No

4. Is the manuscript presented in an intelligible fashion and written in standard English?

Reviewer #1: Yes

Reviewer #2: Yes

5. Review Comments to the Author

Reviewer #1: The authors propose a study aimed at investigating the influence of homeostatic state on RPE signals. They manipulated homeostatic state by varying the blood glucose levels with two paradigms, one ascending (start low and grow) and the other descending (start high and decrease). The results indicate a clear modulation RPE x glucose level in several cortical and subcortical areas, including the VTA, PBN, and caudate nucleus. The homeostatic modulation of RPE was found to depend also on the glucose level phase (ascending or descending). I found the study interesting and very relevant for the understanding of the biological/evolutionary meaning of RPE. I have no major concerns about the methods, nonetheless the authors should improve the clarity of results exposition, in order to make their findings more easily understandable by the reader. In particular (sorted by appearance order in the manuscript):

1. Figure 1b. The authors should define already here as “ascending” and “descending” the two glucose conditions. Moreover, it is not specified the meaning of the eight different time courses in each condition.

2. Figure 2. I suppose the y axis indicates PSC. It should be specified. The authors should stress out the matching between the meaning of the time courses color here and in Figure 1d

3. Figure 3b and Figure 4b are hard to understand. I think the trajectory furthest away from the viewer does not provide important information, it is difficult to read and confounds the reader (it seems an additional trajectory besides those described in Figure 1d). Moreover, the authors should help the reader in matching these trajectories with those described in Figure 1d (who’s who?).

4. The authors should provide a table of fMRI results.

Finally, recent literature on computational neuroscience of Reinforcement Learning (RL) is evidencing how decision-making in the mammalian brain is strongly driven by optimizing the net value (discounted by costs) about both primary and non-primary rewards (e.g. Alexander & Brown 2011; Silvetti, Alexander et al. 2014; Verguts et al. 2015; Silvetti, Vassena et al., 2018). I think this work is relevant for clarifying how reward and RPE signals are dependent from internal states, and how the latter modulate RL processes, suggesting that RPEs are not only the comparison between expectation and environmental outcome (objective), but are dependent on homeostatic states (subjective). These results should be linked to the above literature in the Discussion section, as they contribute to a paradigmatic change that shifts RL-based decision-making from being “objective” to be more “subjective”.

Reviewer #2: Morville and colleagues use FMRI to investigate the relationship between reward prediction error (RPE) signals and homeostatic state (ascending and descending serum glucose trajectories). The authors find that RPE responses in midbrain structures are sensitive to glucose trajectories, which demonstrates a link between RPE responses and homeostatic processes. The authors acknowledge several important limitations in their work, and draw generally appropriate conclusions from their findings. Although I think the manuscript possesses many good features, I have a number of concerns that should be addressed in a revision.

Major Concerns

1) My biggest problem with the manuscript is the lack of methodological details. First, it is difficult to evaluate what was done with respect to the analyses. I think the authors authors rely on the canonical hemodynamic response function. But, it's not clear to me if this shape would capture meaningful variation tied to difference in neuronal activity in this design (cf. Chen et al., 2015). Some discussion of the potential impact of HDR shape would be important to include. Second, it is not clear how the authors modeled RPEs and included those responses as parametric modulators in their analyses. Third, given the design and concatenation of the FMRI sessions, how did this interact with the high pass filter?

2) The authors state that they will release the data on NeuroVault. However, this repository is for statistical maps, which should be shared of course (e.g., Botvinik-Nezer et al., 2020). I recommend that the authors share the raw data (formatted into the Brain Imaging Data Structure) on OpenNeuro (Gorgolewski et al., 2015). With such a rich dataset and many important open questions (e.g., fitting the RL model directly to the brain data), it would be unfortunate not to share the data openly and publicly. (Of course, I realize the authors might still be working on some of these questions, and thus could elect to embargo the data for some period of time. But, as is, the manuscript is not compliant with data sharing policies at this journal and hence why I list this as a "major" comment.)

Minor Concerns

1) Although the paper from Stauffer and colleagues (2014) links dopaminergic activity to marginal utility and the explanation in the Introduction makes sense to me, I think it would be worth explaining this concept further in the Discussion. Unless I missed it, the authors do not mention marginal utility again after the Introduction.

2) I think the concept of "three interacting valuation systems" (Sternson & Eiselt 2017) could get confused with other valuation systems, as described in related work (e.g., Rangel et al., 2008). Please try to clarify and reconcile these different frameworks.

3) The reward cue and liquid would be very difficult to separate without any jitter. In addition. the responses to these events could occur in the same or consecutive TRs. Please clarify how these phases of the task were modeled (see above major comment).

4) How were the scanning parameters optimized to record midbrain responses? Were there any corrections for physiological noise?

5) How did the authors ensure that participants were compliant with the 12-hour fasting protocol?

6) Please check figure legends for completeness/accuracy. Some elements (e.g., line color for the HDR functions in Figure 2 are not defined).

7) In Figure 2, please add error bars to the HDR functions.

8) In Figures 3 and 4, consider using a different color bar for the brain activation and serum glucose levels. As is, the color bars looks somewhat similar, which could lead to confusion.

6. PLOS authors have the option to publish the peer review history of their article (what does this mean?). If published, this will include your full peer review and any attached files.

Reviewer #1: No

Reviewer #2: No

---

## [Author Response · Author response to Decision Letter 0]

10 Aug 2020

Our reviewer responses are uploaded as a separate document in this resubmission. We make use of several screenshots of the revised document in our reply which this text field cannot support. We hope this does not cause inconvenience.

---

## [Editor Report · Decision Letter 1]

19 Aug 2020

PONE-D-20-05797R1

Reward signalling in brainstem nuclei under fluctuating blood glucose

PLOS ONE

Dear Dr. Hulme,

Thank you for submitting your manuscript to PLOS ONE. After careful consideration, we feel that it has merit but does not fully meet PLOS ONE’s publication criteria as it currently stands. Therefore, we invite you to submit a revised version of the manuscript that addresses the points raised during the review process.

I thank the authors for responding to the comments by myself and the reviewers. The paper became much clearer as a result. I would like you to look at these (minor) unclarity issues still.

The study design is not entirely clear: Every subject does 4 sessions, but is it LLHH, or LHHL, or a random order,…?P 6: Ascending and descending are used for the first time here; and only explained on page 10. Please define on first pass.P 6: “A standard fixed effects level…” Do you mean that a fixed effects level model was fitted across all data? Although a fixed effects model in itself is standard, it is definitely not standard to apply it across subjects (without random effects). Please explicitly comment on this in the paper.The model clarification is good, but I wonder about the statement “Since there is no overt behaviour…” This is true, but i don’t see how this is relevant. Behavioral data are needed indeed when you want to estimate the parameters of a model; but you can perfectly well generate regressors for a Rescorla-Wagner, TD, or any other model (which, apparently, is what you do). TD is just about how you model how value “percolates” to earlier time points (such as cue) from reward. You can generate regressors for that model. But the way I see it, your model does not learn, but you give it for each time point (cue, feedback, …), the equilibrium value that it has to converge to, based on the statistics of the experiment? Then say explicitly that your model does not learn, not that you don’t have behavioral data to fit a model.P 12: “that this was true”: change to “this was true”

We look forward to receiving your revised manuscript.

Kind regards,

Tom Verguts

Academic Editor

PLOS ONE

---

## [Author Response · Author response to Decision Letter 1]

17 Nov 2020

Please see the attached file for reviewer comments and our reply.

---

## [Editor Report · Decision Letter 2]

1 Dec 2020

Reward signalling in brainstem nuclei under fluctuating blood glucose

PONE-D-20-05797R2

Dear Dr. Hulme,

We’re pleased to inform you that your manuscript has been judged scientifically suitable for publication and will be formally accepted for publication once it meets all outstanding technical requirements.

Kind regards,

Tom Verguts

Academic Editor

PLOS ONE
---

## [Editor Report · Acceptance letter]

29 Mar 2021

PONE-D-20-05797R2 

Reward signalling in brainstem nuclei under fluctuating blood glucose 

Dear Dr. Hulme:

I'm pleased to inform you that your manuscript has been deemed suitable for publication in PLOS ONE. Congratulations! Your manuscript is now with our production department. 

Kind regards, 

on behalf of

Dr. Tom Verguts 

Academic Editor

PLOS ONE